# Advances in Biomimetic Nanoparticles for Targeted Cancer Therapy and Diagnosis

**DOI:** 10.3390/molecules26165052

**Published:** 2021-08-20

**Authors:** Chaw Yee Beh, Ray Putra Prajnamitra, Li-Lun Chen, Patrick Ching-Ho Hsieh

**Affiliations:** Institute of Biomedical Sciences, Academia Sinica, Taipei 11529, Taiwan; chawyee@ibms.sinica.edu.tw (C.Y.B.); ray.prajnamitra@ibms.sinica.edu.tw (R.P.P.); jeffrey.llc.md10@nycu.edu.tw (L.-L.C.)

**Keywords:** biomimetic, active targeting, nanoparticles, cancer therapy, immunotherapy, imaging

## Abstract

Biomimetic nanoparticles have recently emerged as a novel drug delivery platform to improve drug biocompatibility and specificity at the desired disease site, especially the tumour microenvironment. Conventional nanoparticles often encounter rapid clearance by the immune system and have poor drug-targeting effects. The rapid development of nanotechnology provides an opportunity to integrate different types of biomaterials onto the surface of nanoparticles, which enables them to mimic the natural biological features and functions of the cells. This mimicry strategy favours the escape of biomimetic nanoparticles from clearance by the immune system and reduces potential toxic side effects. Despite the rapid development in this field, not much has progressed to the clinical stage. Thus, there is an urgent need to develop biomimetic-based nanomedicine to produce a highly specific and effective drug delivery system, especially for malignant tumours, which can be used for clinical purposes. Here, the recent developments for various types of biomimetic nanoparticles are discussed, along with their applications for cancer imaging and treatments.

## 1. Introduction

Nanoparticle-based drug delivery systems have been implemented for years. This platform has been widely developed due to its stability, improved drug solubility, high drug payload and ability to allow various routes of administration, especially for parenteral applications [1,2]. Traditionally, nanoparticles are designed to be able to passively target tumour sites by taking advantage of the persistent enhanced permeability and retention (EPR) effect in the tumour environment (Figure 1) [3,4]. In this condition, an elevated increase in vascular endothelial growth factor [5], nitric oxide, peroxynitrite and bradykinin [6] increases the inter-endothelial cell gaps, resulting in increased vascular permeability, which then enables the nanoparticles to permeate more easily into the tumour site. Furthermore, the lack of lymphatic drainage in tumours [6,7] ensures that the nanoparticles are retained, resulting in increased accumulation over time. Generally, conventional nanoparticle drug carriers are liposomes, solid lipid nanoparticles, nanostructured lipid carriers, polymers, dendrimers, micelles and magnetic and inorganic nanoparticles. Among them, those made with biodegradable materials induce lower toxicity and are retained longer in the circulatory system [8].

Unfortunately, there are some obstacles due to tumour heterogeneity and the natural defence system of mammalian organisms. It was reported that the intensity of the EPR effect is largely influenced by the heterogeneity of the EPR effect within the tumours, tumour staging and different tumour types [4]. Different tumour types have different vasculatures, which can influence the physiological environment in the tumour such as the vascular structure, the blood flow rate and vascular permeability. In addition, the physicochemical properties of nanoparticles such as their size, shape and surface charges also contribute to the variations in the EPR effect and drug accumulation [9]. This makes it difficult to predict the success of a nanoparticle-based drug delivery system which solely relies on the EPR effect. Additionally, the reticuloendothelial system (RES) is another main obstacle which clears out the nanoparticles from the body once they are recognised as “foreign materials”, leading to a poor therapeutic outcome [8]. Besides the RES system, the off-targeting effect of nanoparticles is also one of the major challenges where nanoparticle accumulation in organs such as the kidneys, heart, lungs or bone marrow may elicit unwanted toxicity, induce undesired immune responses and lower the overall accumulation in the tumour [8].

Biomimetic nanoparticles are an emerging class of nanoparticles whose surface is integrated or fabricated with biomaterials capable of mimicking the biological features and functions of native cells. Thanks to this, biomimetic nanoparticles possess greatly improved biocompatibility, high target specificity, a long retention time and minimal undesired immune responses [10]. There are various types of biomaterials that can be used to coat nanoparticles to enable their biomimicry ability such as cell membranes derived from erythrocytes [11,12], neutrophils [13], natural killer (NK) cells [14], macrophages [15,16], platelets [17], extracellular vesicles [18] and cancer cells [19,20]. There are also nature-inspired biomaterials such as monoclonal antibodies [21], natural proteins [22,23] and viral capsids [24] as well as synthetic biomaterials such as targeting peptides [25] and aptamers [26].

Among these, it has been proved that red blood cell (RBC) membrane-coated nanoparticles greatly improved the half-life, resembling the long circulation time of normal RBCs [11,12]. Moreover, nanoparticles with surface modification using a cancer-derived plasma membrane or natural ligands markedly improve the drug specificity by binding to the specific cellular markers located on the surface of cancer cell membranes. These surface-modified nanoparticles tend to bind to cancer cells of the same origin but not to the surrounding normal cells. This strategy has been applied to the plasma membrane of immune cells such as T cells and macrophages. As an example, it was shown that macrophage membrane-coated biomimetic nanoparticles successfully achieved tumour homing with a sustained drug effect in response to tumour microenvironment stimuli [16]. Cancer cells possess vastly different cellular membrane proteins or markers which distinguish them from normal cells, resulting in tumour-homing capability. For the same reason, different types of cancer can also be distinguished from one another. It has been shown that nanoparticles coated with tumour cells from various origins can home in on their specific tumour type in vivo through the ability known as homotypic targeting [27]. Furthermore, these biomimetic nanoparticles can avoid detection by the immune system and favourably accumulate at the desired disease sites.

Overall, this review focuses on the recent advancements of biomimetic nanoparticles engineered with different biomaterials. We will describe their role and ability to interact with the biological environment in the body, especially in the tumour microenvironment. We will also discuss the application of biomimetic nanoparticles for cancer therapies.

## 2. Design and Fabrication of Biomimetic Nanoparticles

The advancement of nanotechnology has improved the design and fabrication of nanoparticles for effective drug delivery. This technology has also given birth to strategies in which various types of biomimicking materials are fabricated onto the surface of nanoparticles, known as biomimetic nanoparticles. In this review, we have categorised the biomimetic nanoparticles based on three major types which are natural protein-based, targeting ligand and cell membrane-coated biomimetic nanoparticles. Each of these types of nanomaterial is inspired by nature, and each possesses its own advantages and shortcomings. The synthetic biomimicking moieties usually require attachment of the moieties (e.g., through covalent bonding) onto the surface of nanoparticles. Other biomimicking moieties derived directly from nature, such as cell membranes, are directly used to coat the nanoparticles inside their cavity through extrusion of the nanoparticles and the membranes [28].

### 2.1. Natural Protein-Based Biomimetic Nanoparticles

Proteins are one of the essential components in the human body and are involved in most of the cellular processes. They have excellent structural integrity and diverse functions which enable them to be modified or reprogrammed. Owing to their remarkable versatility and biocompatibility [22,23,29], proteins have been extensively studied in targeted drug delivery systems (Figure 2). There are various types of proteins that have been used in nanoparticles such as serum albumin, ferritin, lipoproteins and virus-like particles. As an effective tool in targeted drug delivery systems, studies have revealed the great potential of natural proteins in targeting specificity, pH or stimulus-induced conformational changes to achieve sustained drug effects, drug stability and synergistic effects [30,31]. For example, Abraxane is a well-known albumin-bound nanoparticle of paclitaxel (PTX) for clinical use to combat cancer [32]. Being bound to albumin improves the solubility of the hydrophobic drug.

#### 2.1.1. Serum Albumin-Fabricated Nanoparticle

Serum albumin has long been explored as a versatile natural protein for nanoparticle fabrication. It is the most abundant protein in the blood and has an average half-life of 3 weeks. Its long half-life in the circulation can be attributed to its C-terminal end which is regulated by the neonatal Fc receptor and is able to protect it from intracellular degradation [33]. Studies have shown that many metastatic tumours are overexpressed with albumin-binding proteins such as secreted protein and rich in cysteine (SPARC) and glycoprotein-60 (gp-60) [34]. To achieve biomimetic transport mediated via SPARC and gp-60, the surface of bovine serum albumin (BSA) nanoparticles was fabricated with a cell-penetrating peptide which is a low-molecular weight protamine. The low-molecular weight protamine-coated BSA nanoparticles were then co-encapsulated with PTX and ferentinide, called L-BSA nanoparticles [35]. Through flow cytometry analysis, it was found that L-BSA nanoparticles achieved 2.5-fold higher cellular uptake than BSA nanoparticles. The fabrication of the low-molecular weight protamine on the surface of nanoparticles was aimed to further enhance the blood–brain barrier penetration and intratumoural infiltration. The blood–brain barrier is recognised as a major challenge in developing treatments for brain tumours—its tight architecture separates the systemic circulation from the brain parenchyma and restricts the extravasation of therapeutics and nanoparticles from the circulation and into the brain [36]. Therefore, nanoparticles need to be further modified in order to facilitate penetration through the blood–brain barrier. In the orthotopic brain glioma model, L-BSA nanoparticles showed significant accumulation in the brain within two hours and distribution across the tumour region. L-BSA nanoparticles showed the longest survival time among all treatment groups, which was 37 days, compared to BSA nanoparticles with 33 days. Additionally, L-BSA nanoparticles exerted an anti-angiogenesis effect, indicated by a reduced CD31 level in orthotopic brain glioma.

The formyl peptide receptor, which falls under the G protein-coupled receptor family, has been found overexpressed in malignant tumours such as triple-negative breast cancer [37]. An active targeting approach to the formyl peptide receptor was developed whereby biomimetic human serum albumin (HSA) nanoparticles were loaded with PTX and modified with the W peptide, a formyl peptide receptor ligand, denoted as Wpep-HSA-PTX [38]. Through an in vitro drug release test, it was shown that the HSA-based nanoparticles were able to release their load through a redox-responsive manner. In a triple-negative breast cancer murine model, Wpep-HSA-PTX achieved good pharmacokinetic behaviour with a 7.5-fold higher drug residence time compared to the control PTX. Ex vivo imaging revealed that Wpep-HSA-PTX significantly accumulated in the tumour region, suggesting an improved tumour-homing effect. Furthermore, Wpep-HSA-PTX efficiently reduced the tumour burden by inducing a strong pro-apoptotic effect in a triple-negative breast tumour.

#### 2.1.2. Ferritin Protein Cage

Ferritin (Fn) is a major iron storage protein which can be found in bacteria, plant and animal cells. It consists of 24 protein subunits that will self-assemble into a hollow protein cage and plays a major role in cellular iron homeostasis [39]. Indeed, its natural self-assembly ability, high thermal stability and hollow spherical structure make it a good candidate for biomimetic drug delivery [39]. During the iron uptake process, Fn will specifically bind to transferrin receptor 1, which is overexpressed in most cancer cell types [40]. In a clinical gastric cancer study, the magneto-Fn nanoparticle-based immunohistochemistry method was utilised for the screening of transferrin receptor 1 expression. The study revealed that around 73.03% of human gastric cancer samples were found to be associated with the upregulation of transferrin receptor 1. Therefore, a doxorubicin (DOX)-loaded heavy chain Fn nanocarrier (HFn-DOX) was developed to target transferrin receptor 1 for the treatment of gastric cancer [41]. Using a gastric cancer patient xenograft mouse model, the tumour growth inhibition rate of the HFn-DOX-treated group was reported as 91.1%, with a median survival of 55 days, while for the free DOX group, the rate was only 41.6%, with a median survival of 27 days. This proved that the Fn carrier can not only be used as a biomimetic drug carrier but also as a useful prognosis marker in human gastric cancer.

Previous studies have shown that the combination of disulfiram and copper complex is highly effective against various cancers [42,43,44]. Due to a lack of tumour specificity, albumin crossed-linked with human Fn loaded with copper ion nanoparticles was synthesised [45]. The stability of the nanoparticles was measured, and it was revealed that copper ions would only be released under an acidic environment, which indicated their stability in the blood circulation. It was shown that the nanoparticles achieved a longer half-life, and there was a significant in vivo accumulation in 4T1 triple-negative breast tumour-bearing mice. The combination treatment with disulfiram, which was administered orally, produced a strong anti-tumour effect with no toxicity, such as neurotoxicity. With this combination strategy, these nanoparticles can potentially serve as a biocompatible anti-tumour agent.

Small interfering RNAs (siRNAs) have long been shown to be able to selectively silence gene expression, especially in cancer therapy [46]. Pediconi et al. [47] synthesised a piperazine-based compound conjugated to chimeric archaeal ferritin loaded with siRNA, called siRNA-Pas-HumAfFt nanoparticles. Due to the nature of the negatively charged siRNA, the incorporation of piperazine-based compounds into chimeric archaeal ferritin was conducted to increase the positive charges which would promote spontaneous binding with siRNA molecules. The nanoparticles would selectively bind to transferrin receptor 1 and then sequentially deliver the siRNA into HeLa, HepG2 and MCF-7 cancer cells with a significant silencing effect towards glyceraldehyde-3-phosphate dehydrogenase, which, in turn, resulted in a reduction in cancer cell glycolysis and decreased proliferation.

#### 2.1.3. Virus-Like Particles

Another natural protein cage that has been studied extensively is virus-like particles, defined as a virus protein shell without a virus genome. Most of the fabricated protein cages are derived from virus-like particles, and they have been shown to have excellent stability in vivo [48]. Among all these natural protein-based nanoparticles, virus-like particles have a major advantage in their well-programmed infection capability, targeting their specific host cells and entering any cells they want to infect [48]. For example, hepatitis B virus has the ability to specifically bind to human hepatic cells with its surface antigen L protein [49]. Genetically modified hepatitis B core particles have been successfully developed and fabricated with affinity ligands called Z_HER2_ affibodies. Z_HER2_ affibody-coated hepatitis B core particles possess a targeting ability to human epidermal growth factor receptor-2 (HER2)-expressed cancer cells: both in in vitro and in vivo experiments, specific uptake of Z_HER2_ affibody-coated hepatitis B core particles in a HER2-positive tumour was shown [50]. Shan et al. [51] reported another application of hepatitis B virus by successfully encapsulating DOX into hepatitis B core particles without any structural modifications and modified the surface with a tumour-targeting peptide, the lipophilic NS5A peptide. It was shown that the nanoparticles were able to target integrin α_v_β_3_ and exerted an excellent anti-tumour effect with reduced cardiotoxicity of DOX in B16F10 tumour-bearing mice.

Integrins play an essential role in cell–cell communications. It is well known that these receptors are overexpressed in most cancers and can act as an entry platform for viruses and bacteria into the host cells for infection [52]. The natural host-infecting ability of foot-and-mouth disease virus occurs through binding with an integrin receptor via a conserved arginine-glycine-aspartic acid motif and enters the host cells through endocytosis [53]. Yan et al. [54] conjugated the surface of foot-and-mouth disease virus-like particles with DOX through covalent bonds and studied the anti-tumour effect. It was shown that the nanoparticles induced HeLa cells’ apoptosis in vitro and significantly reduced the tumour burden in a HeLa-based xenograft nude mouse model with minimised DOX side effects.

Using a neurotropic virus to deliver drugs which target brain tumours through the blood–brain barrier, rabies virus glycoprotein-amplified hierarchical targeted hybrid particles with a magnetolytic effect were developed and were dual loaded with two tumour-penetrating drugs in which DOX was loaded on boron-doped graphene quantum dots (B-GQDs/DOX) and palbociclib was loaded on pH-responsive dendrimers/palbociclib (pH-Den/Palbociclib) [55]. The hybrid nanoparticles successfully penetrated through the blood–brain barrier via spinal cord transportation and resided at the target site under a weakly acidic tumour environment. In response to the magnetic field, the hybrid nanoparticles disassembled into single B-GQDs/DOX and pH-Den/Palbociclib and penetrated deeply into the brain tumour. Both DOX and palbociclib achieved a synergistic anti-tumour effect and significantly prolonged the survival in an orthotopic brain tumour mouse model.

### 2.2. Nanoparticles with Targeting Ligands

Leveraging the EPR effect to achieve tumour accumulation has been considered successful, even resulting in passive targeting liposomal DOX nanoparticles being the first nanoparticle-based anticancer therapeutics approved by the Food and Drug Administration for clinical use [56]. Unfortunately, a recent meta-analysis study conducted by the Chan group showed that passive targeting results in poor nanoparticle accumulation in the tumour, with a median of 0.6% of the injected dose [57]. In order to increase accumulation in the tumour, and therefore increase therapeutic efficacy, nanoparticles with active targeting moieties are needed. Additionally, in recent years, more studies have shown that the entrance of nanoparticles into the tumour follows a more active approach by the endothelial cells instead of a passive EPR effect-based approach [58]. Hence, more studies have now focused on decorating the surface of nanoparticles with targeting moieties in order to augment the ability of nanoparticles to actively target and accumulate in the tumour (Figure 2).

#### 2.2.1. Folic Acid as a Targeting Ligand

The folate receptor is known to be overexpressed on the surface of various tumour cells, such as breast, lung, kidney and ovarian cancers, and is widely recognised as a tumour biomarker [59]. Its ligand, folic acid, has been utilised as a tumour-targeting molecule, not only because of its high binding affinity towards the folate receptor but also because it is non-immunogenic and has high stability and tissue permeability. Furthermore, the unique chemical structure of folic acid makes it possible for it to be conjugated to other scaffolds without compromising its binding to the folate receptor [59].

Ferroptosis, a term first coined in 2012, is a type of programmed cell death that is driven by iron-dependent peroxidation of phospholipids, known as the Fenton reaction [60]. As with any other radical reactions, the Fenton reaction also goes through the initiation, propagation and termination steps. Fe^2+^ and Fe^3+^ react with intracellular H_2_O_2_ to form reactive oxygen species (ROS) and drive the production of phospholipid hydroperoxides. If these hydroperoxide molecules are not quickly removed by glutathione peroxidase 4 (GPX4), they will promote the propagation of more phospholipid hydroperoxides. Termination of this chain reaction is driven by antioxidants, GPX4 or ferroptosis suppressor protein 1. If the reaction is not stopped, it will affect other parts of the cells, resulting in the breakdown of membrane integrity and, ultimately, cell death [60]. Anticancer therapy based on Fenton and Fenton-like reactions is also known as chemodynamic therapy [61,62]. With this knowledge in mind, a study published in 2020 reported the delivery of GPX4 siRNA and cisplatin loaded in iron oxide nanoparticles for chemodynamic therapy of glioblastoma multiforme (GBM) [63]. The study’s strategy was to induce ferroptosis of the GBM cells using iron oxide nanoparticles, to induce apoptosis and increase the intracellular level of H_2_O_2_ aided by platinum in cisplatin and to enhance ferroptosis by silencing the expression of GPX4 using the siRNA. Folic acid was then incorporated on the nanoparticle surface as a targeting moiety capable of homing to folate receptor-overexpressed GBM cells. An in vitro experiment showed that the nanoparticles were able to induce apoptosis and ferroptosis in U87MG and P3 GBM cells. An in vivo experiment showed that U87MG tumour-bearing mice treated with the nanoparticles had the lowest tumour burden and the longest survival [63].

Another study reported the use of synergistic therapy combining chemodynamic, photodynamic and photothermal therapies for the treatment of breast cancer (Figure 3) [64]. Both photodynamic and photothermal therapies are non-invasive cancer therapies that rely on photosensitisers, which are molecules that can be activated by light of a specific wavelength [65]. In photodynamic therapy, the activated photosensitisers generate ROS, resulting in localised chemical damage, which are able to directly kill tumour cells and damage the tumour vasculature, leading to tumour cell death [66]. In photothermal therapy, the activated photosensitiser absorbs energy from the photons and releases heat, resulting in localised thermal damage and irreversible tissue damage and cell necrosis [65]. Yang and co-workers used FePtMn inorganic nanoparticles anchored with Ce6 as a photosensitiser and decorated them with folic acid as a targeting moiety [64]. With the aid of folic acid, the nanoparticles were able to accumulate efficiently in the tumour. Within the acidic tumour microenvironment, the FePtMn nanoparticles released Fe^2+^ and Mn^2+^ cations. The release of Mn^2+^ aided tumour imaging and diagnosis through enhanced T_1_-weighted magnetic resonance imaging and synergistically improved Fe-induced T_2_-weighted magnetic resonance imaging. Furthermore, the release of Fe^2+^ induced the Fenton reaction, resulting in ferroptosis of the tumour cells. Platinum from the nanoparticles helped to release oxygen, which was subsequently converted into ROS by Ce6 following irradiation with a 660 nm laser. Additionally, upon irradiation with an 808 nm laser, Ce6 exhibited its photothermal therapeutic activity. These cleverly designed nanoparticles successfully alleviated the tumour burden of 4T1 breast tumour-bearing mice [64].

#### 2.2.2. Monoclonal Antibodies (mAbs) as Targeting Ligands

The folate receptor is not the only unique receptor found on the surface of cancer cells. The epidermal growth factor receptors (EGFRs) are a family of structurally related receptor tyrosine kinases expressed in many types of cells belonging to the epithelial, mesenchymal and neuronal lineages [67]. The aberrant tyrosine kinase activity and expression levels of this receptor family, such as EGFR and HER2, have been identified in many types of cancers, including glioma, melanoma and colorectal, oesophageal, lung, breast, prostate, pancreatic and ovarian cancers [67,68,69]. Both EGFR and HER2 have been well studied, and clinical trials have been performed based on targeting these receptors [67]. One of the ways to precisely target them is by using antibodies, and therefore a lot of studies have also made use of these antibodies not as therapeutics per se but, instead, as targeting moieties for nanoparticles.

EGFR is overexpressed in non-small cell lung cancers and colorectal cancers, where it plays a role either in tumorigenesis or tumour growth [68]. Several anti-EGFR monoclonal antibodies have been developed and approved by the Food and Drug Administration such as cetuximab, panitumumab and bevacizumab, among others, for the purpose of targeted therapy [67,69,70]. Cetuximab, in particular, is capable of binding to the extracellular domain of EGFR, inhibiting the binding of EGFR ligands such as epidermal growth factor, thus inhibiting downstream signalling pathways [70]. A lot of studies in recent years have reported the use of cetuximab as a targeting ligand for nanoparticles. One of these studies reported the delivery of poly(β-L-malic acid) nanoparticles containing an antisense oligonucleotide to silence the expression of the protein kinase CK2 catalytic α subunit [71]. Studies have shown that CK2 is overexpressed in GBM, supporting tumour growth by suppressing apoptosis, and 34% of GBMs have an amplified gene expression that encodes CK2α. Therefore, the researchers aimed to inhibit GBM growth by silencing the expression of CK2α. In order to cross the blood–brain barrier, the researchers decorated their nanoparticles with anti-transferrin receptor monoclonal antibodies. To further facilitate targeting to the tumour, anti-EGFR (cetuximab) monoclonal antibodies were also decorated on the nanoparticle surface. Using LN229 GBM cell lines, they observed a decreased expression of the CK2α protein and its downstream effectors such as Akt and c-Myc, which are essential for tumour growth. For their in vivo experiment, LN229 and U87MG GBM-bearing mice were treated with these nanoparticles. Although the researchers did not report a change in tumour size, they noted that both groups of mice had a significantly increased survival rate, with the tumours of LN229 GBM-bearing mice having extensive areas of tissue necrosis, indicating the success of the treatment [71].

HER2 was first identified in breast cancer and thus is most commonly associated with it, even though more recent studies have found it is also implicated in several other cancers such as lung, oesophageal, gastric and bladder cancers [68]. Although breast cancer is heterogenous, around 20–25% of breast cancers worldwide overexpress HER2 [72]. Hence, anti-HER2 monoclonal antibodies such as trastuzumab and pertuzumab have been developed for the purpose of targeted therapy. Trastuzumab exerts its anti-tumour activity through binding to the extracellular domain of HER2, exhibiting antibody-dependent cell-mediated cytotoxicity, inhibiting downstream signal transduction pathways and angiogenesis, inducing cell cycle arrest and apoptosis and interfering with DNA repair [72,73]. Pertuzumab, on the other hand, prevents heterodimerisation of HER2 with HER3, resulting in proliferation inhibition [72]. Even though chemotherapy combined with trastuzumab adjuvant therapy is effective, disease relapse has been reported in some patients [74]. Moreover, resistance to anti-HER2 therapy is also well documented [73,74]. Through a HER2-targeted strategy, Li and co-workers designed a theranostic platform comprising bismuth sulfide@mesoporous silica nanoparticles loaded with DOX and decorated with trastuzumab [21]. The bismuth-based nanoparticles served as a photothermal agent and a drug delivery vehicle carrying DOX, making them capable of promoting synergistic photothermal therapy and chemotherapy towards HER2-positive breast tumours. Furthermore, they were also able to act as computed tomography (CT) contrast imaging agents due to the bismuth-based enhanced X-ray attenuation coefficient, resulting in enhanced CT imaging. An in vivo therapeutic efficacy study using SKBR3 breast tumour-bearing mice revealed that the nanoparticles were able to precisely accumulate in the tumour site, and that the mice receiving nanoparticle-mediated photothermal therapy and chemotherapy were shown to have the lowest tumour burden [21].

#### 2.2.3. Tumour-Penetrating Peptides as Targeting Ligands

One of the early examples of tumour-penetrating peptides was reported by the Ruoslahti group in 2009 [25]. This tumour-penetrating peptide was discovered following ex vivo phage display selection with cells from bone tumours and subsequent in vivo selection for peptides capable of homing to the bone tumours. They found that a cyclic peptide with a CRGDKGPDC sequence, which was then named iRGD, was also able to internalise into prostate cancer cells in vitro. They also found that in vivo, iRGD was able to home selectively to tumours but not to normal tissue using mice bearing orthotopic xenografts of prostate, pancreatic ductal and breast cancer, as well as bone and brain xenografts of prostate carcinoma and others. Furthermore, they found that iRGD-coated Abraxane injected into 22Rv1 human prostate carcinoma-bearing mice was able to accumulate more and penetrate deeper into the tumour, resulting in a significantly lower tumour burden. Additionally, the growth of BT474 human breast tumour cells was also successfully inhibited following treatment with iRGD-coated Abraxane [25].

Further studies revealed that iRGD peptides are able to bind to tumour vessels through endothelial cell-specific α_v_β_3_ and α_v_β_5_ integrins, which open their gateway to the tumour [75]. Following a proteolytic cleavage and binding to a second receptor, the C-end Rule pathway is activated. This pathway mediates the penetration of iRGD-decorated moieties, ranging from small molecules to nanoparticles, into the tumour tissue [75]. One of the major challenges in drug delivery to tumours is poor penetration due to a dense connective tissue stroma (such as in pancreatic cancer) as well as a high interstitial pressure [75]. Therefore, the tumour-penetrating ability endowed by iRGD decoration has opened a pathway in cancer therapy that can potentially improve the efficacy of the already existing cancer therapeutics.

A recent study reported the use of iRGD-decorated nanoparticles for enhanced radiotherapy efficacy towards oesophageal squamous cell carcinoma [76]. Tumour hypoxia has been shown to be one of the reasons behind tumour resistance to radiotherapy. Radiotherapy uses ionising radiation to induce DNA damage in the tumour, in the form of DNA radicals, which may further be exacerbated by oxygen under normoxia conditions, resulting in DNA breakage and tumour cell death. The absence of oxygen in hypoxia prevents DNA breakage and promotes DNA repair, ultimately making the tumour resistant to radiotherapy [77]. Therefore, attenuation of hypoxia may enhance radiotherapy efficacy and prevent resistance. For this study, the researchers used cerium oxide (CeO_2_) nanoparticles known to exhibit excellent nanozyme activities. Nanoenzymes are nanomaterials with enzyme-like properties, and CeO_2_ nanoparticles are known to have catalase activity which can decompose hydrogen peroxide into water and oxygen. Hence, the purpose of using CeO_2_ is to alleviate tumour hypoxia through in situ oxygen generation, thereby promoting radiation-induced DNA damage by preventing its repair and inhibiting tumour growth. Unfortunately, ultrasmall CeO_2_ nanoparticles are prone to aggregation which may potentially reduce their nanozyme activity. Hence, to prevent aggregation and increase their stability, ultrasmall CeO_2_ nanoparticles were anchored to 2D graphdiyne. The nanoparticles were then conjugated with miR-181a micro-RNA, capable of enhancing apoptosis induced by DNA damage by targeting RAD17 and regulating the Chk2 pathway. These nanoplatforms with and without miR-181a were then tested against human ESCC KYSE-30 cells in vitro (under normoxia and hypoxia conditions) and also in vivo. For nanoparticles without miR-181a, in vitro results showed an increased number of apoptotic cells, while in vivo results showed a lower tumour burden. When the miR-181a-conjugated nanoplatform was used, the in vivo study showed an even lower tumour burden in the mice receiving the treatment. Overall, the results showed that the combination of nanoparticle treatment with radiotherapy attenuated hypoxia and enhanced the radiotherapy efficacy, resulting in alleviation of the tumour burden, which could even be further enhanced by conjugation with miR-181a [76].

#### 2.2.4. Aptamers as Targeting Ligands

Aptamers are short single-stranded oligonucleotide molecules (DNA or RNA). They are identified following a meticulous selection process enabling them to specifically bind to certain proteins [78]. Due to their small size and flexible structure, they are able to bind to binding domains that are not accessible by large-sized antibodies. The large size of antibodies may also lower their therapeutic efficacy by hampering tissue penetration [79]. Aptamers are identified in vitro using cell systematic evolution of ligands by exponential enrichment (cell-SELEX) [80]. In this procedure, a pool of a DNA library containing billions of sequences is first incubated with target cells for positive selection. Aptamers of interest are those bound on the cell surface. Following removal of unbound aptamers, aptamers of interest are extracted and incubated with non-target cells for negative selection. The unbound aptamers obtained from this step are collected and amplified by PCR. These steps are repeated for several rounds until a sequence with high affinity towards the target cells and low affinity towards non-target cells is identified [80]. This aptamer, which is cell-specific, can then be used for various purposes, such as a targeting ligand.

Earlier this year, our group published a study using therapeutic-loaded nanoparticles decorated with aptamers which were able to piggyback on the surface of immune cells and used them as a shuttle bus to target the heart following ischaemic injury and the pancreatic tumour site [26]. Both of these unrelated diseases with an entirely different pathophysiology had one aspect in common: following a myocardial ischaemia reperfusion episode and during tumour growth, monocytes are constantly recruited to these sites and, upon extravasation, will differentiate to become macrophages. With this in mind, we developed a strategy where, instead of targeting the injury site directly, we used monocytes to carry our therapeutic-loaded nanoparticles to the injury site, thereby enabling the treatment of two diseases with the same strategy. We used IOX2, an inhibitor of hypoxia-inducible factor-1α, for the treatment of the heart, while gemcitabine was used for the treatment of pancreatic ductal adenocarcinoma. We identified an aptamer, J10, following cell-SELEX using RAW264.7 and J774A.1 murine monocyte/macrophage cell lines for positive selections and an SVEC murine endothelial cell line for negative selection. The drugs were loaded separately into different liposome nanoparticles, and the J10 aptamer was anchored on the surface. We found that the treatment successfully improved heart function after myocardial ischaemia reperfusion and ameliorated the pancreatic tumour burden in mice. The treatment also successfully prolonged the survival of both groups of mice. Furthermore, when we used the nanoplatform to treat mice with liver metastasis, we found that the treatment successfully ameliorated the metastatic tumour burden and also increased their survival [26]. Since our strategy is not disease-specific, we believe that our nanoplatform can be used for the treatment of diseases that involve monocyte recruitment in the pathophysiology.

A lysosome-targeting aptamer has been used to reprogram the lysosome within cytotoxic T cells, creating super-cytotoxic T lymphocytes for cancer immunotherapy [81]. In this research, metal–organic framework (MOF) nanoparticles were synthesised using Zn^2+^ ions connected by 2-methylimidazole bridging units, also known as ZIF-8. The ZIF-8 MOF was loaded with therapeutic proteins perforin and granzyme B and coated with Ca^2+^ ions that can synergistically enhance the anticancer effects of both proteins. The nanoparticles were then coated with a lysosome-targeting aptamer and incubated with activated T cells. Following internalisation by T cells, the nanoparticles would attach to the lysosomes, penetrate inside and release and store their contents within the lysosomes, resulting in super-cytotoxic lysosomes. Once the T cells were further activated by the major histocompatibility complex of tumour cells, they would release the content of the super-cytotoxic lysosomes, resulting in higher anti-tumour activity compared to normal T cells. In vivo assessment using 4T1 tumour-bearing mice revealed that treatment with super-cytotoxic T cells attenuated the tumour burden and significantly prolonged mice survival [81]. This study is unique in that the nanoparticles were used to prime the T cells in vitro before being used for treatment and may potentially be used alongside other existing cancer immunotherapy methods.

### 2.3. Cell Membrane-Coated Nanoparticles

Following injection into the bloodstream, nanoparticles will undergo opsonisation due to the protein in the blood, making them recognisable to the phagocytes. Once recognised, the phagocytes will attempt to clear the nanoparticles from the body through phagocytosis before they can reach the cancer site. This will result in a shorter nanoparticle circulation time and, ultimately, less therapeutic efficacy. Attempts to prolong the circulation time in the body and protect the nanoparticles from the immune cells have been made such as surface modification by the attachment of polyethylene glycol (PEG), which increases hydrophilicity, resulting in longer half-lives. Indeed, coating of nanoparticles with PEG has been very successful and has also led to the development of a number of clinical products. However, it was found that the use of PEG-containing products may result in the release of anti-PEG antibody within the patient’s body, which can accelerate the clearance of the subsequent injections of PEG-containing nanoparticles, decreasing their therapeutic efficacy [82].

Another strategy developed to evade clearance by the immune cells was achieved by decorating nanoparticles with the “marker of self” CD47 [83]. This transmembrane glycoprotein is expressed on the surface of many cells, such as RBCs, and is known to interact with signal regulatory protein-α (SIRPα, also known as CD172a) on phagocytes. The CD47–SIRPα interaction results in a “do not eat me” signalling pathway which negatively regulates phagocytosis, preventing CD47-expressing cells from being cleared out by the immune cells. Unfortunately, chemistry-based bioconjugation techniques of CD47 on nanoparticle surfaces may lead to protein denaturation [12]. Furthermore, due to its large size and the complexity of a correct protein folding crucial for its function, CD47-mimicking peptides have been generated and used to decorate nanoparticle surfaces [84]. However, synthesis of these peptides is delicate since they must be designed so that the interaction with SIRPα is not too strong and the signalling is not too weak [83].

Due to the challenges faced in the attempts to enable a nanoparticle immune evasion ability, a biomimicry strategy was developed where, instead of specific surface proteins, nanoparticles were coated with cell membranes as a whole. The cell membranes contain all of the necessary surface proteins that the nanoparticles need for immune evasion and, sometimes, active targeting (Figure 2). Various types of cell membranes such as RBCs, immune cells, platelets and cancer cells, to name a few, have been used to coat nanoparticles. While inherent tumour-targeting capability is present in some cell types such as immune cells, this may not necessarily be the case in other cells. Therefore, a lot of research has also reported nanoparticles coated with cell membranes to enable immune evasion which were also further decorated with active targeting moieties to enable tumour targeting.

#### 2.3.1. RBC Membrane-Coated Nanoparticles

This type of cell membrane-coated nanoparticle is one of the first membrane-coated nanoparticles invented. As nature’s main carrier of oxygen in the body, RBCs are not recognised by the immune system as a foreign body and thus can evade immune clearance. One of the reasons for this ability is due to the presence of CD47 on their surface [85]. A proof-of-concept study was reported in 2011 where poly(lactic-co-glycolic) acid (PLGA) nanoparticles coated with either RBC membranes or PEG were compared. The results indicated that the RBC membrane-coated nanoparticles had a considerably longer circulation time (~40 h) compared to the PEG-decorated nanoparticles (~16 h) in vivo, showing that the nanoparticles managed to evade clearance by the immune system better [12]. Since then, the use of RBC membrane-coated nanoparticles has grown rapidly in the field of targeted drug delivery.

A clever use of RBC membrane-coated nanoparticles for combined photodynamic and chemotherapy treatment of HeLa tumour-bearing nude mice has been reported [86]. For the cargoes, the researchers used the PTX dimer as a chemotherapy prodrug and 5,10,15,20-tetraphenylchlorin as the photosensitiser that can generate singlet oxygen species. To generate the nanoparticles, they encapsulated the cargoes into methoxypoly(ethylene glycol)-block-poly(D,L-lactide) and coated them with RBC membranes. Following uptake by the tumour through the EPR effect, light was irradiated, and the photosensitiser generated ROS that killed the tumour cells. Interestingly, the ROS reacted with the PTX dimer prodrug molecules, cleaved them and generated paclitaxel, which further aided in killing the tumour cells. As a result, the mice treated with the combined chemotherapy and photodynamic therapy using RBC membrane-coated nanoparticles had the smallest tumours compared to those only subjected to either chemotherapy or photodynamic therapy only. Furthermore, the RBC membrane coating also helped the nanoparticles to prolong their time in the circulation and to improve their accumulation in the tumour site [86].

Another strategy using RBC membrane-coated nanoparticles to relieve tumour hypoxia has also been reported [87]. It is common for solid tumours to exhibit hypoxia due to dysregulated angiogenesis and poor blood perfusion. Aiming to relieve hypoxia, the researchers took advantage of perfluorocarbon, which is a type of organic molecule that not only is able to dissolve oxygen with excellent solubility but is also inert and nontoxic. They encapsulated perfluorocarbon inside PLGA nanoparticle cores and coated them with RBC membranes. Administration of the nanoparticles into 4T1 breast tumour-bearing mice successfully relieved hypoxia in the tumour as soon as eight hours after the injection, as assessed by tumour histology. Mice with and without nanoparticle pre-treatment were subsequently subjected to radiotherapy 24 h after injection. The researchers found that pre-treatment with their nanoparticles improved the radiotherapy efficacy as it was able to suppress the tumour growth better than radiotherapy alone [87]. This example shows how combined therapies using membrane coating with radiotherapy can further increase the therapeutic efficacy.

#### 2.3.2. Immune Cell Membrane-Coated Nanoparticles

White blood cells (leukocytes) are another major constituent of the blood cells. They help to defend the host from pathogens, and they consist of immune cells such as monocytes, macrophages, neutrophils, NK cells, dendritic cells and T and B cells. Since one of the main purposes of coating nanoparticles with cell membranes is to avoid clearance by the immune cells, disguising the nanoparticles as part of the immune system by using their membrane has given birth to novel therapeutics and new treatment methods in cancer therapy. Furthermore, immune cells such as macrophages and neutrophils have been implicated in cancer, giving them the ability to home in on cancer sites [88,89]. This allows nanoparticles coated with these cell membranes to possess an active cancer-targeting capability, something that cannot be achieved using RBC membranes.

A recent study revealed that during lung metastasis of breast cancer, macrophages are able to use α4-integrin on their surface to bind with the vascular cell adhesion molecule-1 (VCAM-1) on the surface of the cancer cells. This interaction triggers Akt survival signalling which then promotes their survival and metastatic capability [90]. To leverage the α4-integrin–VCAM-1 interaction, Cao et al. [91] encapsulated the cytotoxic anticancer drug emtansine in liposome nanoparticles and coated them with macrophage membranes isolated from the RAW264.7 murine monocyte/macrophage cell line in order to generate macrophage membrane-coated nanoparticles used to target 4T1 metastatic foci in the lungs. As controls, membrane-coated nanoparticles with blocked α4-integrin (blocked nanoparticles) and non-coated nanoparticles were used. Ex vivo imaging showed both normal and blocked nanoparticles to have a prolonged circulation time compared to non-coated nanoparticles owing to the macrophage membrane coating. Furthermore, analysis showed that the membrane-coated nanoparticles were able to target the lungs more efficiently compared to the other groups and also had higher nanoparticle accumulation in the lungs, reaching a maximum at four hours post-injection. Accumulation in the metastatic foci and internalisation into the metastatic cancer cells were confirmed using confocal microscopy, which clearly showed higher accumulation and internalisation for the membrane-coated nanoparticles. A therapeutic efficacy assessment was then performed, and it was found that the lungs of the mice treated by the membrane-coated nanoparticles had the fewest number of metastatic nodules, indicating the lowest metastatic burden [91]. Overall, this study serves as a proof of concept wherein macrophage membranes can be exploited for targeted cancer therapy due to the specific interaction between α4-integrin and VCAM-1.

Neutrophils are the most abundant cells that make up the innate immune system in the blood. As a professional phagocyte, one of their main roles is to defend against infecting pathogens. Neutrophils have been implicated in cancer metastasis [89], particularly in the formation of the premetastatic niche. Cancer cells disseminate from the primary tumour into the circulation and adhere to certain distant organs. These cells are known as circulating tumour cells (CTCs) [92]. In order for CTCs to adhere to and proliferate in the metastatic site, the site would have to first be primed, forming the premetastatic niche, through a series of inflammatory cascades. As the result of this event, neutrophils with the ability to inhibit tumour-suppressive CD8^+^ T cells are recruited to the niche, supporting the metastasis formation [93]. Additionally, neutrophils have also been reported to interact with CTCs in the circulation. With this knowledge in hand, Kang et al. [94] skilfully engineered neutrophil membrane-coated nanoparticles that were able to target CTCs, the premetastatic niche and established metastatic foci. The researchers used carfilzomib, a proteasome inhibitor, in PLGA coated with a pre-activated neutrophil membrane in a mouse model of 4T1 triple-negative breast cancer with lung metastasis. Through in vivo flow cytometry and confocal intravital microscopy, it was found that their nanoplatform was able to capture the CTCs more efficiently compared to non-coated nanoparticles. The neutrophil membrane-coated nanoparticles were able to eliminate the CTCs (~23% apoptotic 4T1 cells) more efficiently than the non-coated nanoparticles (~2%). Simultaneously, the membrane-coated nanoparticles were able to accumulate in the premetastatic niche and reduced the formation of metastatic nodules. Furthermore, through quantification of S100A9 in the lung, which corresponds to the likelihood of premetastatic niche formation, they found that membrane-coated nanoparticles were able to reduce the amount of S100A9^+^ neutrophils (77%) better compared to non-coated nanoparticles (44%). Lastly, in vivo and ex vivo bioluminescence and near-infrared imaging showed that the neutrophil membrane-coated nanoparticles were able to accumulate more in the metastatic foci (~2-fold) and inhibit their growth better (~87% reduction) compared to the non-coated nanoparticles (~43% reduction). Further analysis by TUNNEL staining showed that treatment with the membrane-coated nanoparticles induced the highest rate of 4T1 cell apoptosis (~84%) compared to non-coated nanoparticles (~41%) or the free carfilzomib drug (~20%) [94]. Overall, this shows that the neutrophil membrane-coated nanoparticles were able to successfully alleviate the metastatic burden, not only by eliminating the CTCs but also by preventing the formation of the premetastatic niche and reducing the already established metastatic burden.

NK cells are also a part of the innate immune system and the first line of defence against cancer and infection [95]. Coating NK cell membranes onto nanoparticles can be conducted to acquire the unique properties of NK cells. Unlike T cells, they are able to recognise and spontaneously eliminate infected cells or tumour cells without prior activation [96]. They are also able to activate T cells to aid in killing tumour cells by stimulating the maturation of antigen-presenting cells such as dendritic cells [97]. Furthermore, their surface proteins are also able to induce pro-inflammatory M1 macrophage polarisation, resulting in an anti-tumour immune response. Deng et al. [97] used photodynamic therapy for their choice of treatment by loading 4,4′,4′′,4′′′-(porphine-5,10,15,20-tetrayl)tetrakis(benzoic acid) into polymeric nanoparticles and coated them with NK cell membranes. Near-infrared fluorescent imaging in vivo and ex vivo showed significantly higher accumulation of the membrane-coated nanoparticles compared to the non-coated nanoparticles. In a mouse model of a bilateral subcutaneous 4T1 tumour, the treatment managed to prolong the nanoparticle circulation time, reduce the tumour burden in the mice, both in the primary and distant tumour sites, and successfully prolong the survival of the mice. Furthermore, using flow cytometry, they showed a considerable increase in M1 macrophages, mature dendritic cells and effector T cells (CD4^+^ and CD8^+^) in the tumours of the mice treated by the membrane-coated nanoparticles. These results showed that photodynamic therapy combined with the unique inherent property of NK cell membranes managed to augment the anti-tumour activity through the generation of ROS by the photosensitiser as well as a stronger immune response that involves M1 macrophage polarisation and increased infiltration of effector T cells. This effect was observed not only in the primary tumour site but also in the distant site, which indicates that this treatment produced an abscopal effect [97].

T cells are a component of the immune system that plays an important role in the adaptive immune response. In cancers, cytotoxic T lymphocytes are capable of recognising tumour antigens presented on major histocompatibility complex molecules, which can activate them to kill the tumour cells [98]. Even though T cell-based therapies have shown remarkable outcomes in the clinic, the immunosuppressive nature of the tumour microenvironment may cause the T cells to be exhausted, losing their anti-tumour efficacy [99]. Alternatively, therapies using immune checkpoint blockades, such as anti-PD-1 or anti-PD-L1, are able to prevent T cell exhaustion by inhibiting PD-1–PD-L1 interaction [100]. Although this strategy has shown good outcomes, immune-related adverse side effects have also been observed, due to the excessively activated immune system [101]. In an endeavour to overcome these shortcomings, Kang et al. [102] developed T cell membrane-coated nanoparticles which were envisioned to be able to target the tumour (through LFA-1 adhesion protein) and eradicate the tumour cells by mimicking the anti-tumour activity of T cells. Since the T cell membrane-coated nanoparticles were not live cells, they did not respond to the immunosuppressive nature of the tumour microenvironment and therefore did not experience exhaustion and were unaffected by the immune checkpoints. On the contrary, the T cell membrane-coated nanoparticles were able to act as a T cell decoy by blocking the immune checkpoint interaction between T cells and the cancer cells, as well as scavenging the immunosuppressive molecules, allowing T cells to keep their cytotoxic functions towards cancer cells. To achieve this, anticancer drug-loaded PLGA nanoparticles were coated with T cell membranes derived from the EL4 cell line. Flow cytometry confirmed the presence of important T cell membrane proteins such as LFA-1, TGF-β1R and PD-1. For control groups, the researchers used trypsin-treated nanoparticles and LFA1-blocked membrane-coated nanoparticles as well as anti-PD-L1 antibody. The cancer models used for this study were a subcutaneous B16F10 melanoma model, melanoma with lung metastasis and a subcutaneous Lewis lung cancer model. Ex vivo imaging showed that nanoparticle accumulation in the tumour was significantly higher for the membrane-coated nanoparticles compared to other groups. Their therapeutic efficacy was also superior, even compared to anti-PD-L1, because they were able to prevent the growth of B16F10 tumour cells due to their ability to target tumour cells. When loaded with dacarbazine, the performance of the nanoparticles was even better compared to empty nanoparticles, by showing an increase in tumour-killing efficacy with the lowest amount of lung metastatic nodules. They were also able to prevent the tumour growth of the subcutaneous tumour-bearing mice and significantly increased their survival. Finally, analysis of various T cell populations in the tumour following various treatments showed the highest amount of cells in the T cell membrane-coated nanoparticle-treated tumour, which indicated that the treatment prevented T cell exhaustion [102]. Overall, therapy using the nanoparticles developed in this study was able to overcome the shortcomings of adoptive T cell transfer therapy and anti-PD-L1 therapies. Since these nanoparticles can be prepared from T cell lines, they can potentially be used as a low-cost and less laborious but better alternative to adoptive T cell transfer therapy.

#### 2.3.3. Platelet Membrane-Coated Nanoparticles

Platelets are anucleate blood cells known to play a key role in haemostasis and maintaining the vascular wall from any injuries, preventing blood loss [103]. They have also been implicated in blood vessel development by assisting the recruitment of progenitor cells to the vasculature, thus promoting angiogenesis. Platelets also play an important role in cancers. An excessive platelet count of more than 450,000 per microlitre (also known as thrombocytosis) is a hallmark of many cancers and is usually associated with poor prognosis [104,105]. The tumour endothelium secretes pro-angiogenic factors during angiogenesis which also results in the recruitment of platelets. Upon reaching the tumour site, platelets release more angiogenic and permeability factors, further promoting vascular remodelling, inflammation and a leaky vasculature [104]. CTCs in the blood stream are capable of interacting with leukocytes and platelets, forming aggregates that are able to attach to the endothelium, implicating the role of platelets in cancer metastasis [105]. Interestingly, platelet homeostasis is regulated by their CD47 expression, which also prevents their clearance by the immune system [106]. Therefore, using platelet membranes as a coating material for nanoparticles will endow them with not only cancer targeting but also immune evasion capabilities.

Another work published earlier this year leveraged the P-selectin–CD44 interaction between platelets and tumour cells to design tumour-targeting platelet membrane-coated nanoparticles for multimodal triple-negative breast cancer imaging and combined therapy [107]. Nanocarbons were used for photoacoustic imaging as well as photothermal therapy. For the chemotherapy drug, DOX was chosen. Additionally, perfluoropentane was used for ultrasonic imaging and to enhance the therapeutic efficacy of DOX by enhancing tumour vascular cell permeability and promoting drug release at the tumour site. Nanocarbons, DOX and perfluoropentane were encapsulated in PLGA nanoparticles and subsequently coated with platelet membranes, which enabled them to evade clearance by the immune system and to home in on the tumour site. The combination of nanocarbons and perfluoropentane endowed the nanoparticles with photoacoustic and ultrasonic imaging capabilities, improving the diagnosis. Furthermore, this combination therapy produced a synergistic effect and managed to completely eliminate the tumour. Finally, biosafety analysis by histology of major organs did not reveal organ damage, indicating the safety of this nanoplatform [107].

#### 2.3.4. Cancer Cell Membrane-Coated Nanoparticles

The advantage of using cancer cell membranes is their inherent properties, such as the ability to evade the immune system [108] and to bind to each other (membrane-coated pic aggregation/binding) [109]. A proof-of-concept study was reported in 2014 where a cancer cell membrane derived from B16F10 melanoma cells was coated onto PLGA nanoparticles [19]. When the resulting cancer cell membrane nanoparticles were used to deliver tumour-associated antigens, the researchers observed dendritic cell maturation as well as T cell activation. Furthermore, the researchers tested the homotypic binding ability by comparing human MDA-MB-435 breast cancer cell membrane-coated nanoparticles with RBC membrane-coated nanoparticles. They observed that the uptake of cancer cell membrane nanoparticles by the cancer cells was 40-fold higher than that of RBC membrane-coated nanoparticles [19]. All of these results have established cancer cell membrane-coated nanoparticles as novel biomimetic nanoparticles with broad potential usage such as for drug delivery, imaging and cancer vaccines.

One study reporting the use of cancer cell membrane-coated nanoparticles as a cancer vaccine was published in 2019 [110]. Surgical removal of tumours is the main choice for tumour treatment; however, due to incomplete removal, residual tumour cells may grow again, resulting in relapse. In this study, the authors used 4T1 triple-negative breast cancer, B16F10 melanoma and 4T1 with lung metastasis mouse models. Tumour membrane cells (4T1 or B16F10) were harvested from the tumour cells at the primary site following surgical removal and coated onto black phosphorus quantum dots. The resulting membrane-coated nanoparticles were then loaded into a thermosensitive hydrogel which also carries granulocyte-macrophage colony-stimulating factor and lipopolysaccharide. Following injection of nanoparticles, the release of the colony-stimulating factor resulted in the recruitment of dendritic cells, which also recognised the tumour antigen, whilst the release of lipopolysaccharide promoted dendritic cell maturation. They would then present the tumour antigen to enable T cell activation, subsequently killing any residual or metastatic tumour cells. The mice were then subjected to near-infrared radiation which caused the quantum dots to release heat locally. The heat helped to enhance dendritic cell functions and increase vascular permeability, resulting in improved accumulation in the tumour site. Furthermore, combination treatment with anti-PD-1 successfully prevented T cell exhaustion. Treatment with the cancer cell membrane-coated nanoparticles managed to prevent tumour recurrence in both 4T1 and B16F10 mouse models and also prevented metastasis to the lungs. Ultimately, the mice survival rate was dramatically increased following the treatment [110]. Therefore, this platform can potentially be used for personalised tumour vaccines in the future, where a patient’s surgically removed tumour-derived cellular membranes will be used as a nanoparticle coating, thereby preventing clearance by the immune system.

In a different application, a study using upconversion nanoparticles coated with cancer cell membranes reported yielding highly specific tumour imaging by taking advantage of the homotypic binding property of cancer cell membrane-coated nanoparticles [27]. Upconversion nanoparticles are a type of inorganic nanoparticle which typically contains lanthanide-based transition metals such as Yb^3+^, Tm^3+^, Er^3+^ or Ho^3+^. These nanoparticles, upon excitation by near-infrared light (low-energy photons), are able to produce an emission with a shorter wavelength (blue or green, high-energy photons) [111]. The authors of this study used PEGylated β-NaYF_4_:Er^3+^,Yb^3+^ upconversion nanoparticles coated with cancer cell membranes derived from MDA-MB-435 human breast cancer cells (MDA nanoparticles), DU 145 human prostate cancer cells (DU nanoparticles), CAL 27 human squamous cancer cells (CAL nanoparticles) and HCT 116 human colorectal cancer cells (HCT nanoparticles). The researchers also used RBC membrane-coated nanoparticles, non-coated nanoparticles and folic acid-decorated nanoparticles as controls. Irradiation with near-infrared resulted in the upconversion nanoparticles emitting a green emission, which enabled upconversion luminescence imaging. To investigate immune evasion and targeting abilities, all nanoparticles were administered into MDA-MB-435 tumour-bearing mice. In vivo and ex vivo imaging revealed that all cell membrane-coated nanoparticles showed higher accumulation in the tumour and less in the RES, indicating immune evasion ability endowed by the cell membrane coating. Furthermore, MDA nanoparticles were shown to have the highest accumulation in the tumour site compared to all of the other nanoparticles. On the other hand, DU, CAL, HCT and RBC membrane-coated nanoparticles showed similar accumulation. All of these findings indicated that these cancer cell membrane-coated nanoparticles possessed an immune evasion ability similar to that of RBC membrane-coated nanoparticles; however, the best result was still achieved when the type of cell membrane used for coating matched the type of tumour in the host due to homotypic binding preference [27].

#### 2.3.5. Exosome Membrane-Coated Nanoparticles

Exosomes are extracellular vesicles secreted by all cell types which are also responsible for various biological and cellular functions including intercell communications [112]. They are found in bodily fluids and are released as a result of cell activation or changes in the physiological environment [113]. The cargoes that exosomes carry can be diverse and may contain proteins, nucleic acids (RNA and DNA) and lipids, all of which are encapsulated by a lipid bilayer membrane. The exosome membrane is known to express CD47, which provides exosomes with their immune evasion ability [114]. Exosomes have been known to play a role in cancers (diagnosis, prognosis and metastasis), cardiovascular diseases (coronary artery disease and heart failure), pregnancy disorders and even organ transplantation [115]. The exosomes secreted by tumour cells have been shown to aid tumour growth by influencing other cells, creating a tumour-supporting microenvironment. They may influence other cells to support neoangiogenesis and also protect the tumour cells from phagocytes by suppressing the immune system. In cancer metastasis, exosomes have been shown to exhibit organ tropism, depending on the type of cancer from which they originate. They also help to form the premetastatic niche by leaving the primary tumour site through the circulation and arriving at a certain distant secondary organ, where they create a tumour-supporting microenvironment suitable for metastasis [113]. For example, breast cancer cell-secreted exosomes possess excellent lung-homing capabilities due to α_6_β_4_ and α_6_β_1_ integrins and liver-homing capabilities due to α_v_β_5_ integrin [116]. All of these reasons have made exosomes a unique source of membranes for nanoparticle coatings.

Another study reported the use of exosome-coated nanoparticles to deliver siRNA to suppress lung metastasis following removal of primary breast tumours [117]. S100A4 is a protein known to promote tumour progression and metastasis. The authors of this study aimed to silence the expression of this protein in the lungs using S100A4 siRNA, thereby preventing lung metastasis. Cationic BSA conjugated to the siRNA was generated and conjugated with exosome membranes secreted by 4T1 breast cancer cells. After the resulting membrane-coated nanoparticles had been injected into non-tumour bearing mice, the authors observed the highest nanoparticle accumulation in the lungs but not in the other organs, confirming the organ tropism properties of breast cancer-secreted exosomes. A postoperative 4T1 lung metastasis mouse model was then developed by removing the primary tumour surgically, followed by an intravenous injection of 4T1 cells to induce metastasis. Following injection of nanoparticles, the researchers observed that mice receiving the membrane-coated nanoparticles had the lowest metastatic burden, indicating that the treatment had successfully suppressed lung metastasis. Furthermore, analysis of S100A4 expression in the lungs after treatment also showed that mice receiving the membrane-coated nanoparticles had very low S100A4 expression, indicating that their expression had been silenced [117].

In order to increase the tumour-targeting ability of the nanoparticles, one study decorated its exosome membrane-coated nanoparticles with an additional targeting moiety [118]. In this study, the authors leveraged the overexpression of the mesenchymal–epithelial transition in triple-negative breast cancer by using its binding peptide as a targeting moiety. To generate the nanoplatform, DOX-loaded PLGA was coated with RAW264.7 macrophage-derived exosomes to form the membrane-coated nanoparticles, which were then further decorated with the binding peptide. In vitro study using the MDA-MB-231 triple-negative breast cancer cell line showed that the membrane-coated nanoparticles had excellent accumulation in the cancer cells; however, the peptide-decorated membrane-coated nanoparticles exhibited the highest accumulation. TUNEL assay showed that tumour cells treated with exosome membrane-coated nanoparticles had a high apoptotic rate, but when they were treated with the nanoparticles with the additional peptide targeting moiety, the apoptotic rate was even higher. A similar trend was observed in vivo, where accumulation in the tumour was the highest for peptide-decorated membrane-coated nanoparticles, resulting in the smallest tumour burden [118]. Overall, this study showed that although coating with macrophage-derived exosome membranes was able to endow the nanoparticles with tumour-targeting capability, it can be further enhanced by adding a targeting moiety, resulting in a higher therapeutic efficacy.

## 3. Application of Biomimetic Nanoparticles in Cancer Diagnostics and Therapy

Recent development in nanotechnology has resulted in the emergence of biomimetic nanoparticles which possess several advantages over the conventional nanoparticles. They have great improvements in the delivery of chemotherapy drugs, such as the ability to penetrate deep into the tumour microenvironment and to avoid the RES while, at the same time, having minimal side effects and high target specificity. Besides being a biomimetic drug carrier in targeted cancer therapy, they can also be adapted for use as a carrier to deliver cancer imaging materials for early cancer diagnosis. Owing to their high biocompatibility, biomimetic nanoparticles can be further developed into anticancer nanovaccines in which they will boost the host immune system and kill the tumour cells in an antigen-specific manner.

### 3.1. Cancer Imaging

Optical imaging systems play an essential role in cancer imaging. It is important to detect cancers in their early stage to ensure prolonged survival and a high recovery rate. Among the conventional diagnostic methods, X-ray diffraction, X-ray CT, positron emission tomography (PET), optical fluorescent light imaging (FLI) and magnetic resonance imaging (MRI) are commonly used nowadays. However, all of these diagnostic methods cannot detect the tumour specifically, especially in early cancer metastasis where they tend to have a poor prognosis and possible misinterpretation [119,120]. Generally, the conventional imaging systems only operate at a short wavelength such as UV ultraviolet and visible light irradiation which tends to have poor tissue penetration, high autofluorescence and invasive photo-damage to tissues [121].

In recent years, the implementation of the near-infrared region (NIR) window in optical imaging systems has significantly improved the current imaging methods due to its ability to penetrate into deeper tissue levels with precise imaging [1]. Compared to the NIR-I window that falls between 700 and 900 nm, the NIR-II window falls between 1000 and 1700 nm, which can significantly improve tissue penetration and, therefore, the imaging depth and sensitivity, with a substantial reduction in tissue autofluorescence [122]. Rare earth-doped nanoparticles (RENPs) have gained considerable attention in NIR-II imaging due to their long fluorescence lifetime, narrow emission spectra, low toxicity and minimal autofluorescence [123]. A study integrated a human triple-negative MDA-MB-231 cell-derived membrane onto PEGylated-RENPs to form cancer cell membrane-coated RENPs for cancer imaging. The success of the membrane coating was proved by the presence of cancer membrane markers such as EGFR, E-cadherin and Na^+^/K^+^-ATPase on the nanoparticles [124]. This study showed that membrane-coated RENPs achieved a two-fold higher tumour site-to-normal ratio compared to PEGylated-RENPs after 24 h post-injection and reduced clearance by the host immune system. Under the NIR-II tumour imaging window, membrane-coated RENPs precisely emitted their signal at the tumour region and aided the complete tumour resection process, which resulted in the absence of residual tumour, as indicated through H&E staining.

Rhodium (Rh)-based nanoparticles have recently been studied for tumour photodynamic therapy, especially in NIR, due to their promising catalase-like activity [125]. In another study, a Rh-based core shell nanostructure was coated with MDA-MB-231 cell-derived membranes and loaded with the photosensitiser indocyanine green to form biomimetic bimetallic nanoparticles for treating hypoxia tumours. This novel nanotherapeutic agent exhibited excellent tumour accumulation, good biocompatibility, bimodal imaging and enhanced photodynamic therapy efficacy. The nanoparticles enhanced the therapeutic effect by converting the tumour microenvironment hydrogen peroxide into oxygen and, subsequently, with the aid of the photosensitiser indocyanine green, significantly elevated the production of tumour-toxic singlet oxygen. Simultaneously, the nanoparticles induced a stronger fluorescence signal under NIR imaging in MDA-MB-231 tumour-bearing BALB/c nude mice after 12 h post-injection.

Apart from NIR fluorescent cancer imaging, gadolinium-diethylenetriaminepentaacetic acid (Gd-DTPA)-based MRI is another extensively used cancer imaging technique in clinics. Gd-DTPA has a rapid clearance rate through glomerular filtration, and therefore a high dose is required for high-resolution imaging, with a toxicity risk associated with it such as nephrogenic systemic fibrosis [126,127]. To address all of these problems, Cai et al. [128] developed HFn-based nanoparticles conjugated with Gd-DTPA that can naturally bind to transferrin receptor 1, which is overexpressed in tumours. The Gd-DTPA-conjugated HFn nanoparticles achieved a T_1_-MR signal intensity value of 4.78 mM^−1^S^−1^, whereas the commercial Gd-DTPA only exceeded 3.97 mM^−1^S^−1^. In human pancreatic and breast cancer xenograft models, the resulting HFn-conjugated gadolinium-based nanoparticles and commercial Gd-DTPA were injected intravenously to evaluate their performance as MRI contrast agents. It was found that a single intravenous injection with only a dosage of 0.016 mmol Gd/kg body weight achieved a significantly enhanced signal that lasted for 30 min post-injection, compared to Gd-DTPA, which only lasted for approximately 10 min. Under magnetic resonance microscopy, this dose of nanoparticles showed a significant improvement in the signal-to-noise ratio which lasted for 60 min. On the other hand, there was no detectable changes in Gd-DTPA, which suggested that Gd-DTPA was not able to detect the tumour in such a low dose. Moreover, MRI revealed that most of the nanoparticles were cleared through the renal system which reduced the possible harmful effects of Gd.

### 3.2. Cancer Immunotherapy

Our immune system is composed of a complex network of immune cells and physical barriers to fight against infection. Tumorigenesis first starts with chronic inflammation accompanied by genomic instability, epigenetic modification and immune evasion [129]. Extensive efforts proved that neoplastic cells have developed a strategy that can avoid the recognition by the host immune system and create an immunosuppressive tumour microenvironment through the recruitment of inflammatory cells and pro-inflammatory cytokines [130]. Since most tumours have low immunogenicity, it is a great challenge for the host immune system to correctly identify the neoplastic cells. Cancer vaccines are one of the strategies in cancer immunotherapy which mainly leverage the host’s adaptive immune system against specific tumour antigens to kill the tumour cells [131]. However, tumour heterogenicity within the tumour is another major obstacle in cancer immunotherapy which leads to a lack of tumour specificity and ineffective therapeutic efficacy [132]. Through nanotechnology, such as biomimetic nanoparticles, it is now possible to modulate the immunosuppressive tumour microenvironment by specific binding or depleting the immunogenic cells which can directly activate the anticancer immune response [133]. Compared to the conventional cancer vaccines, biomimetic nanoparticles, such as cell membrane-derived biomimetic nanoparticles, provide a low immunogenicity, high targeting ability and minimum toxicity which can further enhance the immune response in a specific manner with a long-term immunity [134].

One example of a cancer vaccine utilising cancer cell membrane-coated nanoparticles was reported by Gan et al. [20], who used CpG oligonucleotide-loaded aluminium phosphate nanoparticles coated with a cancer cell membrane derived from the B16F10 murine melanoma cell line. They aimed to use the membrane-coated nanoparticles as a vaccine that is able to target the lymph nodes—the place where T cells and antigen-presenting cells, such as dendritic cells, reside and interact—and improve the host’s immunity against cancers. They examined the prophylactic efficacy of their nanoplatform by vaccinating the mice once a week for three weeks before subcutaneous injection of B16F10 cells. Measurement of tumour burden following cell inoculation revealed the excellent prophylactic efficacy of the vaccine, as the mice treated with the vaccine had the lowest tumour burden. The researchers also examined the therapeutic efficacy of the vaccine by injecting the mice with the membrane-coated nanoparticles after tumour cell inoculation. Measurement of tumour burden also revealed excellent therapeutic efficacy. In both assessments, treatment with cancer membrane-coated nanoparticles successfully increased the survival of the mice [20].

Virus-like particles also hold a great potential that can be exploited and engineered for efficient activation of the tumour microenvironment in the immune system. Kines et al. [24] developed a human papillomavirus 16 virus-like particle (HPV-VLP) conjugated with a photoactivatable drug, IRDye-700, called belzupacap sarotalocan (AU-011). It is known that the mechanism of HPV entrance to the cells is by binding to the heparan sulphate proteoglycan [135]. It is a cell membrane receptor which has been shown to actively participate in tumorigenesis [136,137]. In vitro data showed that AU-011 exhibited cytotoxicity towards murine lung cancer cells after exposure to NIR in a dose-dependent manner. Simultaneously, the caspase-1 activity was elevated after the tumour cell cytotoxicity effect, which indicated immunogenic cell death and potential induction of an adaptive anti-tumour immune response. The results were further validated in a tumour-bearing mouse model, where an increase in the CD45^+^ cell population was observed in the AU-011 + NIR-treated group, which resulted in around 50% of the mice achieving long-term tumour regression. Through the combination of AU-011 and immune checkpoint inhibitors, the therapeutic efficacy was enhanced, in which 70–100% of tumour-bearing animals showed a complete response rate up to 100 days after the treatment. Those mice were then rechallenged with tumour cells, and long-lasting anti-tumour protection was observed. It can be concluded that AU-011 particles have a direct cytotoxic effect on tumour cells accompanied by long-term anti-tumour immunity, and that the effect is further enhanced when combined with checkpoint inhibitors.

Dendritic cell-based immunotherapy has been extensively studied and is known to activate the tumour antigen-specific cytotoxic T lymphocytes to kill cancer cells [138]. Unfortunately, most of the dendritic cell-based immunotherapies do not achieve significant efficacy due to the lack of tumour-specific antigens, availability of major histocompatibility complex molecules and tumour-associated antigens [139]. To overcome these limitations, Ma et al. [140] incorporated different cell membranes onto the surface of PLGA nanoparticles to generate two types of fusion cell membrane nanoparticles. The first fusion cell membrane was composed of murine colon adenocarcinoma MC38 cells and murine dendritic cells from the bone marrow. It was found that the MC-38 fusion cell membrane-coated nanoparticles accumulated twice as high in the spleen and lymph nodes with a significant T cell response, proliferation and activation. Healthy C57BL/6 mice that were vaccinated with the MC-38 fusion cell membrane-coated nanoparticles for two consecutive weeks before MC-38 tumour cell inoculation revealed to have benefitted from significant anti-tumour protection with enhanced infiltration of CD4^+^ and CD8^+^ T cells in the tumour. Similarly, the same approach has been applied to treat a murine glioma GL261 model using GL-261 fusion cell membrane-coated nanoparticles. The results consistently showed significant anti-tumour protection with effective T cell activation and greatly improved the survival days to over 100 days, compared to the 55 days for the control group. This study suggests a new direction of cancer immunotherapy using a biomimetic nanoparticle-based strategy in cancer nanovaccine development.

In addition, Chen et al. [141] very recently reported the use of tumour-associated macrophages (TAMs), instead of regular macrophages, as a source of cell membranes for the treatment of 4T1 triple-negative breast cancer mice. Circulating monocytes extravasate to the tumour site and mature into TAMs with the help of the macrophage colony-stimulating factor 1 secreted by cancer cells [88]. TAMs, as with regular macrophages, can be categorised into the M1 (pro-inflammatory and anti-tumour) and M2 (anti-inflammatory and pro-tumour) phenotypes [142]. Polarisation of TAMs from M1 to M2 is the result of a signalling cascade initiated by the interaction between the colony-stimulating factor 1 and its receptor on the TAM membrane [141]. In this study, a combination of photodynamic therapy and upconversion nanoparticles was used. The authors envisioned that their nanoparticles would not only be able to kill the tumour cells through photodynamic therapy but would also be able to prevent the polarisation of TAMs to M2 by depleting all of the available stimulating factors within the tumour microenvironment due to the presence of its receptor on the surface of their nanoparticles. The upconversion nanoparticles, NaYF_4_:Yb,Er@NaYF_4_, were synthesised and conjugated with Rose Bengal as the photosensitiser for photodynamic therapy purposes. Upon irradiation with a 980 nm laser, the upconversion nanoparticles emitted green (~540 nm) and red (~660 nm) fluorescence, which, in turn, was absorbed by Rose Bengal, releasing ROS and killing the tumour cells. The researchers noticed that their TAM membrane-coated nanoparticles were not phagocytised by macrophages, indicating their ability to evade the immune clearance. They also found that the targeting of the TAM membrane-coated nanoparticles to the tumour was better than that of regular macrophage membrane-coated nanoparticles, which they attributed to the homing effect of TAMs. In vivo examination of the nanoparticles, in comparison to other control groups, revealed their superior ability to prolong the survival of the mice and alleviate the tumour burden in the primary, distant and metastatic sites (lungs), which indicated that the abscopal effect was produced through this treatment. With a reduction in the stimulating factor level, this prevented the polarisation of TAMs to M2 and consequently preserved their ability to attack the tumour, helping to reduce the tumour burden overall [141].

## 4. Perspective View and Concluding Remarks

The biomimetic nanoparticle platform has provided a new insight in the development of targeted cancer drug delivery systems. In this review, different strategies of targeted drug delivery were discussed based on different biomimetic vectors derived from natural proteins, antibodies, various sources of cell membranes and natural ligands. These biomimetic vectors offer a great improvement in targeting specificity, a long circulation time, immune system evasion and versatility, especially in the tumour microenvironment. Besides targeted drug therapy, the application of biomimetic vectors also covers optical imaging. Precise imaging is essential for cancer diagnostics, most importantly during the early stage of cancer, as well as for premetastasis detection. Due to tumour immunogenicity, it is difficult for the host immune system to elicit a potent response to kill the tumour. The incorporation of biomimetic materials into cancer vaccines shows a potential breakthrough in cancer immunotherapy by accurately eliciting the immune system or regulating the tumour microenvironment in the immune system. Although they have been actively explored in preclinical and clinical research, the complexity of biomaterials may pose several challenges for clinical translation. Significant work is required for a detailed understanding of their application in cancer therapeutics by mainly focusing on immunology, biodistribution and therapeutic mechanisms. Future research should investigate these aspects in more detail so that clinical translation can be achieved.

## Figures and Tables

**Figure 1 molecules-26-05052-f001:**
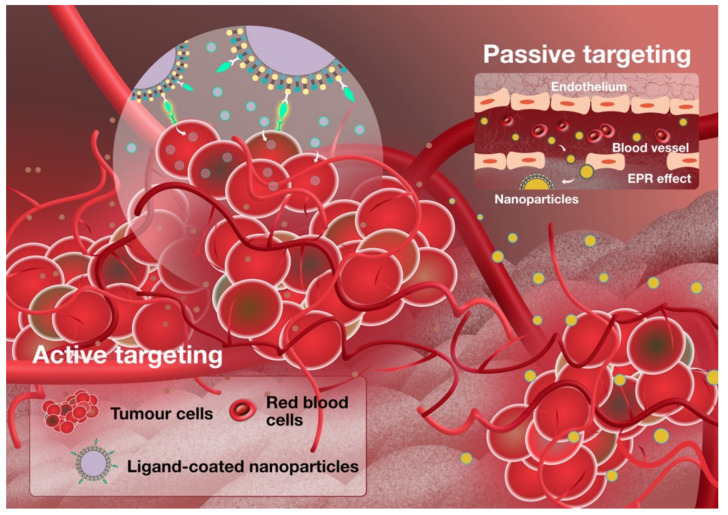
Schematic mechanism of passive targeting via the EPR effect and active targeting through ligand-mediated endocytosis. In passive targeting through the EPR effect, nanoparticles pass through the leaky tumour vasculature and accumulate in the tumour site. In active targeting, ligand-mediated uptake of ligand-coated nanoparticles occurs via interaction between the ligand and specific receptors on the target cells, such as tumour cells, resulting in accumulation in the tumour site.

**Figure 2 molecules-26-05052-f002:**
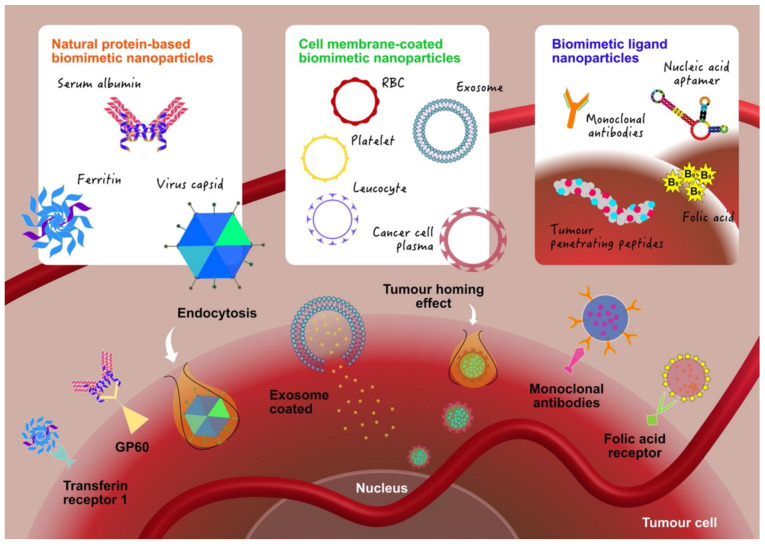
Targeted cell entry strategies of biomimetic nanoparticles based on different types of biomaterials. Tumour cells possess specific receptors that allow biomimetic ligand- or natural protein-coated nanoparticles to bind and achieve active targeting. Cell membrane- or exosome-coated nanoparticles mimic the natural properties and behaviour of the original cells which enables the tumour-homing effect, immune evasion and favourable accumulation in the tumour microenvironment.

**Figure 3 molecules-26-05052-f003:**
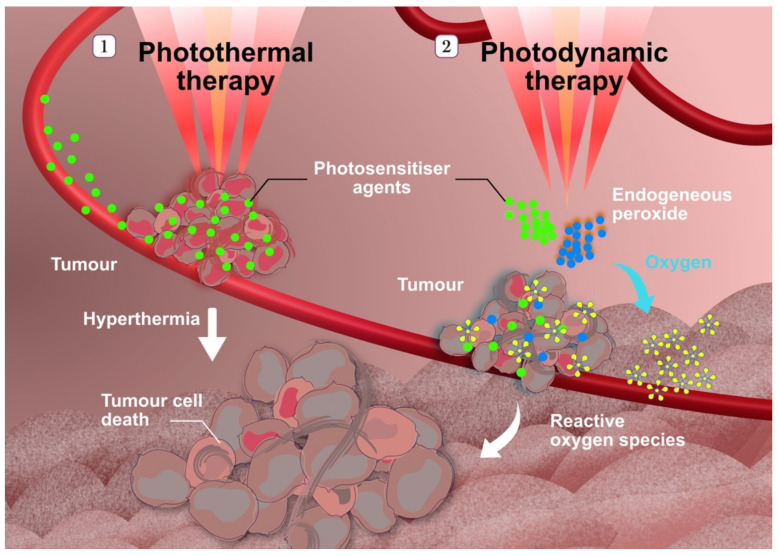
Schematic illustration of light-activated photosensitiser-based therapy in cancer. 1. Laser irradiation induces the photosensitisers to generate heat locally and elevate the temperature at the tumour site, resulting in irreversible tissue damage and cancer cell death. 2. In photodynamic therapy, laser irradiation induces the photosensitisers to convert tumour endogenous peroxide into oxygen and allow it to exit to produce singlet oxygen, a reactive oxygen species, which is tumour-toxic and will directly kill the cancer cells.

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
