# Peer review of "Advances in Biomimetic Nanoparticles for Targeted Cancer Therapy and Diagnosis"

_molecules, 2021, doi:10.3390/molecules26165052_

Round 1

Reviewer 1 Report

Hsieh et al review recent advances in biomimetic nanoparticles for targeted cancer therapy and diagnosis. This article is important for graduate students in this field. However, a major revision is necessary for better presentation.

 I suggests authors present this review in different way as follows.

  1. Introduction
  2. Design and fabrication of biomimetic nanoparticles.

     2.1 Natural Protein-based Biomimetic Nanoparticles

     2.1.1 Serum Albumin Fabricated Nanoparticle

     2.1. 2. Ferritin Protein Cage

     2.1. 3. Virus-Like Particles

2.2. Targeting Ligand-modified Nanoparticles

2.3. Cell Membrane-Coated Nanoparticles

3. Application of Biomimetic Nanoparticles in Cancer Diagnostic and Therapy

4.Perspective View and Concluding Remarks

Author Response

Reviewer 1 Comments

Hsieh et al review recent advances in biomimetic nanoparticles for targeted cancer therapy and diagnosis. This article is important for graduate students in this field. However, a major revision is necessary for better presentation.

I suggests authors present this review in different way as follows.

  1. Introduction
  2. Design and fabrication of biomimetic nanoparticles.

    2.1. Natural Protein-based Biomimetic Nanoparticles

2.1.1. Serum Albumin Fabricated Nanoparticle

2.1.2. Ferritin Protein Cage

2.1.3. Virus-Like Particles

2.2. Targeting Ligand-modified Nanoparticles

2.3. Cell Membrane-Coated Nanoparticles

  1. Application of Biomimetic Nanoparticles in Cancer Diagnostic and Therapy
  2. Perspective View and Concluding Remarks

Reply to Reviewer #1 comment 1

We would like to thank you for taking the time to review our manuscript and for the suggestion. We have now made the appropriate adjustments based on your suggestion. A short introduction in the part of “Design and fabrication of biomimetic nanoparticles” has been included which introduces the subtopics and briefly discussed the differences between the types of nanoparticles in the following sub-subtopics. In all of the sub-subtopics, different strategies to design or fabricate the biomimetic nanoparticles are also discussed including their targeting ability into the tumour microenvironment as well.

We have added the following paragraph for subtopic 2 (page 3, line 94):

  1. Design and Fabrication of Biomimetic Nanoparticles

The advancement of nanotechnology has improved the design and fabrication of nanoparticles for the effective drug delivery. This technology has also given birth to strategies in which various types of biomimicking materials are fabricated onto the surface of nanoparticles, known as biomimetic nanoparticles. In this review, we have categorised the biomimetic nanoparticles based on three major types which are natural protein-based, targeting ligand and cell membrane-coated biomimetic nanoparticles. Each of this type of nanomaterials is inspired from nature and each possesses its own advantages and shortcomings. The synthetic biomimicking moieties usually require attachment of the moieties (e.g., through covalent bonding) onto the surface of nanoparticles. Other bio-mimicking moieties derived directly from nature, such as cell membranes, are directly used to coat the nanoparticles inside their cavity through extrusion of nanoparticles and the membranes [28].

Reviewer 2 Report

In this manuscript by Beh et al, the authors summarize recent advances in biomimetic nanoparticles for cancer therapy and diagnosis. Different strategies to prepare biomimetic nanoparticles are discussed, and applications such as imaging and immunotherapy are covered. The review will be a valuable reference for researchers in this field and I recommend publication of the manuscript. The following minor issues should be addressed before the acceptance of the manuscript.

  1. Some figures from the references can be included in the manuscript to demonstrate the strategies more clearly.
  2. On line 895, the definition of the NIR-II window should be given.

Author Response

Reviewer 2 Comments

In this manuscript by Beh et al, the authors summarize recent advances in biomimetic nanoparticles for cancer therapy and diagnosis. Different strategies to prepare biomimetic nanoparticles are discussed, and applications such as imaging and immunotherapy are covered. The review will be a valuable reference for researchers in this field and I recommend publication of the manuscript. The following minor issues should be addressed before the acceptance of the manuscript.

  1. Some figures from the references can be included in the manuscript to demonstrate the strategies more clearly.
  2. On line 895, the definition of the NIR-II window should be given

Reply to Reviewer #2 comment 1

We would like to thank you for taking the time to review our manuscript and for the suggestion. We have considered this and we think that using published figures from our cited reference is forbidden without the permission of the respective journals. As a substitute, different types and strategies of biomimetic nanoparticles have been discussed in the main text in a detailed way and have been illustrated in our Figure 2. Also, the strategies of receptor-mediated endocytosis, plasma membrane diffusion and tumour homing effects have been demonstrated in Figure 2 as well. If there are any specific points or references that needs to be emphasised, we will be able to revise those parts.

Reply to Reviewer #2 comment 2

On page 20, line 906, we have defined and further introduced the NIR-II window in the tissue optical imaging. We have added the following sentences:

Compared to NIR-I window that falls between 700 and 900 nm, the NIR-II window falls between 1000 and 1700 nm, which can significantly improve tissue penetration and therefore, the imaging depth and sensitivity with substantial reduction of tissue auto-fluorescence [122].

Round 2

Reviewer 1 Report

Authors have improved the manuscript according to my suggestions, and thus I recommend it publication.